# Opioid Addiction and Opioid Receptor Dimerization: Structural Modeling of the OPRD1 and OPRM1 Heterodimer and Its Signaling Pathways

**DOI:** 10.3390/ijms221910290

**Published:** 2021-09-24

**Authors:** Bohua Wu, William Hand, Emil Alexov

**Affiliations:** 1Department of Physics, College of Science, Clemson University, Clemson, SC 29634, USA; bohua@g.clemson.edu; 2Department of Anesthesiology and Perioperative Medicine, Prisma Health, Greenville, SC 29605, USA; William.Hand@prismahealth.org

**Keywords:** opioid addiction, opioid proteins, structural modeling, mutations, energy calculations

## Abstract

Opioid addiction is a complex phenomenon with genetic, social, and other components. Due to such complexity, it is difficult to interpret the outcome of clinical studies, and thus, mutations found in individuals with these addictions are still not indisputably classified as opioid addiction-causing variants. Here, we computationally investigated two such mutations, A6V and N40D, found in the mu opioid receptor gene *OPRM1*. The mutations are located in the extracellular domain of the corresponding protein, which is important to the hetero-dimerization of OPRM1 with the delta opioid receptor protein (OPRD1). The hetero-dimerization of OPRD1–OPRM1 affects the signaling pathways activated by opioids and natural peptides and, thus, could be considered a factor contributing to addiction. In this study, we built four 3D structures of molecular pathways, including the G-protein signaling pathway and the β-arrestin signaling pathway of the heterodimer of OPRD1–OPRM1. We also analyzed the effect of mutations of A6V and N40D on the stability of individual OPRM1/OPRD1 molecules and the OPRD1–OPRM1 heterodimer with the goal of inferring their plausible linkage with opioid addiction. It was found that both mutations slightly destabilize OPRM1/OPRD1 monomers and weaken their association. Since hetero-dimerization is a key step for signaling processes, it is anticipated that both mutations may be causing increased addiction risk.

## 1. Introduction

Opioids are widely used for pain management; however, the number of individuals that are affected with opioid tolerance, dependency, or addiction increased in the past two decades [1,2,3,4]. It is understood that opioid dependence and addiction are complex phenomena stemming from various social and psychological factors [5,6]. The major risk of addiction after taking opioid pain medications is that opioids could induce euphoria in addition to analgesia, which leads to increased risk of addiction [7]. However, emerging evidence indicates that there is also a genetic predisposition to addiction [8,9,10,11]. Currently many genes, including *OPRM1*, *OPRD1*, *DRD2*, *BDNF*, *APBB2*, *KCNG2*, *KCNC1*, *CNIH3*, *RGMA*, *DRD3*, *DRD4*, and *NRXN3* were reported to be associated with addiction [10,12,13]. Recent studies show that around 5–10% of opioid-naïve patients became persistent users after a single low-risk surgery with subsequent opioid-based analgesia [14,15,16,17].

Opioid medications inhibit pain transmission in neurons and in the central nervous system by binding to opioid receptor proteins [18,19,20]. There are three classical types of opioid receptors: mu coded by *OPRM1* gene, delta coded by *OPRD1* gene, and kappa receptors. Once the opioid receptors are activated, they activate one of the corresponding signaling pathways, either the G-protein complex or the β-arrestin [21]. Depending on the ligand binding to the pocket of opioid receptors, either the G-protein signaling pathway or β-arrestin signaling pathway is activated. The binding of agonist activates the G-protein signaling pathway while the binding of antagonist switches the G-protein signaling pathway to the β-arrestin signaling pathway. Opioid receptors were demonstrated to form homodimer or heterodimers [21,22]. Thus, which signaling pathway will be activated depends on the ligands bound to each receptor [23,24]. This indicates the complexity of pathway activation and the role of hetero-dimerization. Below, we focus on the role of hetero-dimerization and review available experimental findings with regards to the signaling pathway being activated.

Studies revealed that OPRM1 and OPRD1 typically form a heterodimer, and the heterodimer formation changes the opioid signaling pathway compared with the signaling pathway activated by the OPRM1 and OPRD1 receptors alone [25]. Previous research showed that, when norepinephrine interacts with OPRD1 or morphine interacts with OPRM1, they trigger the G-protein signaling pathway [26]. The pathway is changed to β-arrestin signaling if morphine and norepinephrine bind onto the OPRD1–OPRM1 heterodimer [26]. In contrast, when oxymorphone and naltrindole bind onto OPRD1–OPRM1 heterodimer, the G-protein signaling pathway is enhanced compared with the G-protein signaling pathway introduced by oxymorphone bound to OPRM1 or naltrindole bound to OPRD1 individually [27]. These observations indicate that the hetero-dimerization of OPRD1–OPRM1 has pronounced physiological effects. Due to that, significant efforts were invested to target hetero-dimerization [28,29].

Mutations in OPRM1 and OPRD1 genes found in the general population were suggested to affect the normal cellular signaling pathway [30,31]. Currently, 29 mutations in the *OPRM1* gene and 10 mutations in *OPRD1* gene are reported in the ClinVar database [32]. Most of them are intron, non-coding transcripts or synonymous variants and only two are missense mutations. Among them, rs1799971 (N40D), rs1799972(A6V), rs510769, and rs2236861 have been implicated in opioid addiction [31,33,34]. Only rs1799971 and rs1799972 are missense mutations (both in the *OPRM1* gene). There are controversial interpretations on the effect of these mutations [35,36]: some studies do not strongly support their link with addiction, while other studies suggested that they are linked with opioid addiction [36,37,38]. Perhaps such conflicting conclusions are due to the multifactorial nature of opioid addiction, likely stemming from small contributions from several alleles [39,40].

It was shown that the conformation change of the N-terminal region of OPRM1 is associated with the activation of signaling pathway [41]. The mutations A6V and N40D are located at the N-terminal region OPRM1 protein and thus are expected to affect conformational flexibility and the selection of the signaling pathway. The mutation A6V was shown to decrease the effects of morphine, buprenorphine, fentanyl, and endogenous opioids and decreased signaling compared with wild-type OPRM1 [31]. The other missense mutation, the N40D mutation, was suggested to play a role in addiction because it has an impact on the signaling transmission that may alter the PKA and pERK1/2 regulation [42]. Both mutations were found to reduce mRNA and OPRM1 levels in cells [43].

The interplay between the hetero-dimerization of OPRD1–OPRM1, the effects of A6V and N40D, and the activation of G-proteins or β-arrestin signaling pathways is not understood. This interplay may be essential for understanding the genetic component of opioid addiction. To facilitate understanding the effects of these missense mutations, we built four 3D structures of molecular pathways, including G-protein signaling and β-arrestin signaling pathways of the heterodimer of OPRD1–OPRM1. For each signaling pathway, we predicted two binding positions of either G-protein or β-arrestin. Furthermore, we conducted a computational investigation of the effect of two missense mutations, A6V and N40D, on the stability of OPRM1 and the binding free energy of the OPRD1–OPRM1 complex. It is speculated that both mutations may affect OPRM1 stability and OPRD1–OPRM1 affinity, affecting the wild-type heterodimer and the corresponding pathway and, thus, contributing to opioid addiction. It is possible that understanding the mechanism and role of specific genetic mutations could lead to patient-specific opioid prescriptions, if a clinician is capable of testing for the abovementioned missense mutations in OPRM1.

## 2. Results and Discussion

### 2.1. 3D Structure Modeling of OPRM1 and OPRD1 Monomers

The 3D structure of the OPRM1 protein is not available in the Protein Data Bank (PDB) [44], while the 3D structure of the OPRD1 protein is available but is fused with a bifunctional peptide. Thus, the sequences of human OPRM1 and OPRD1 proteins were downloaded from Uniprot (P35372 and P41143, correspondingly) [45]. Then, the full-length structures of those proteins were predicted with the I-TASSER server [46]. Five models for each sequence were predicted by the I-TASSER server. The models with the highest confidence of OPRM1 and OPRD1 were chosen for this study. More details are provided in the Appendix A.

### 2.2. 3D Model of Transmembrane OPRD1–OPRM1 Heterodimer

Residues 336–372 of OPRD1 and residues 355–400 of OPRM1 are intrinsically disordered regions. They were removed from the monomer structures before building a model of the OPRD1–OPRM1 heterodimer. The binding model was predicted by ZDOCK [47]. Two thousand models were generated by ZDOCK. The transmembrane segments of OPRD1 and OPRM1 were evaluated by the OPM database [48]. Previous studies have shown that a favorable interface is formed between transmembrane domains of 1 and 7 on OPRM1 with domains 4 and 5 on OPRD1 [49]. Therefore, the selection criteria were as follows: (1) The transmembrane segments predicted by the OPM database could be embedded into the membrane. (2) Cytoplasm and extracellular segments must be in the correct direction. (3) The interaction interface must be between transmembrane domains 1 and 7 on OPRM1 with domains 4 and 5 on OPRD1. Based on these criteria, the best model was selected.

### 2.3. 3D Model of OPRD1–OPRM1 Extracellular Domain Complex

In this study, we constructed a 3D model of the OPRD1–OPRM1 extracellular domain heterodimer using two docking algorithms: the ClusPro server [50] and the ZDOCK server [47]. The structures of the extracellular part on the N-terminal of OPRM1 and OPRD1 were downloaded from the I-TASSER server as explained above. The best models with the highest C-score were used to build the complex. The top ten models were downloaded from each server. Then, we attempted to link the extracellular domains with the model of the transmembrane OPRD1–OPRM1 complex. The extracellular model that provided the best link to the transmembrane OPRD1–OPRM1 complex was selected and linked, and thus, the full-length OPRD1–OPRM1 complex was generated.

### 2.4. 3D Model of Full-Length OPRD1–OPRM1 Heterodimer within Lipid Bilayer

The protein was embedded into a POPC bilayer using the CHARMM-GUI website [51]. The transmembrane segments were determined by the OPM database [48]. When the protein complex was placed into the membrane, the z axis orientation of the complex was aligned with the z axis of the membrane. The whole system of the protein complex was solvated with 0.15 M KCl.

### 2.5. 3D Modeling of the G-Protein and β-Arrestin Coupled with OPRD1–OPRM1 Heterodimer

Here, we built four 3D models of the OPRD1–OPRM1 heterodimer: (1) a G-protein bound to OPRD1 of the heterodimer (Figure 1A), (2) a G-protein bound to OPRM1 of the heterodimer (Figure 1B), (3) a β-arrestin bound to OPRD1 of the heterodimer (Figure 1C), and (4) a β-arrestin bound to OPRM1 of the heterodimer (Figure 1D). These models are generated to probe the possibility of different binding modes of both G-protein and β-arrestin to the OPRD1–OPRM1 heterodimer. Docking was achieved without a lipid membrane; however, after docking, the binding poses were analyzed to remove all poses that interfered with the membrane. Thus, only two out of ten predictions of the G-protein bound to the OPRD1–OPRM1 heterodimer satisfy this requirement. One prediction is the G-protein bound to OPRM1, and another is bound to OPRD1. The binding modes are quite similar. Similarly, applying the same protocol for β-arrestin, two binding modes are generated: one binds OPRD1 to another OPRM1 within the OPRD1–OPRM1 heterodimer. It can be speculated that the preference of the binding of the G-protein and β-arrestin to either OPRD1 or OPRM1 depends on the ligand that activates the process. To provide some insights for this complex process, below, we outline available data taken from the literature.

Each OPRD1 and OPRM1 receptor can bind opioids or endogenous peptides alone and activate either the G-protein signaling or β-arrestin signaling pathways [49,52,53] (Figure 2). For example, β-endorphin activates the G-protein signaling pathway when it binds to the OPRD1 or OPRM1 receptor [21] while enkephalin actives the β-arrestin signaling pathway by binding to the OPRM1 receptor [54]. However, little is known about the activation pathways in the case of the OPRD1–OPRM1 heterodimer and what are the mechanisms causing activation of the G-protein or β-arrestin pathways. Currently, four ligand-mediated signaling pathways are assumed in the case of the OPRD1–OPRM1 heterodimer (Figure 3) [55,56]. When the G-protein-based ligand binds to OPRD1 or OPRM1, it will activate the G-protein signaling pathway [55,56]. When an agonist binds to OPRM1, it will activate the β-arrestin signaling pathway [55,56]. However, when an OPRD1 ligand binds to an OPRD1 protein within the OPRD1–OPRM1 heterodimer and, at the same time, an agonist is bound to an OPRM1 protein, the signaling pathway switches from the β-arrestin signaling pathway to the classic G-protein signaling pathway [55].

### 2.6. Folding Free Energy Change Due to a Mutation

Table 1 shows the results of the folding free energy change due to the mutations calculated with the computational algorithms described in Section 3. In the case of the N40D mutation, there is a consensus, and all methods predict that the mutation destabilizes the extracellular domain of OPRM1. The average magnitude of the folding free energy change is more than half of kcal/mol, which may be a significant factor that contributes to the change in the functionality of OPRM1 and in the selection of the signaling pathway. In the case of the A6V mutation, there is a discrepancy among the methods, such that Duet [57] and CUPSAT [58] disagree with other methods predictions (Table 1). The A6V mutation may destabilize the extracellular structure of OPRM1. Overall, the predicted folding free energy change due to A6V is much smaller than for N40D, and no clear conclusion can be made.

### 2.7. Binding Free Energy Change Due to Mutation

Table 2 shows the results of binding free energy change due to the mutations by different binding methods. In these calculations, only the extracellular domains of OPRD1 and OPRM1 were considered. The predictions made with different methods agree with each other, and they all predicted slight destabilization of the heterodimer. The magnitude of predicted binding free energy change is small; however, it still may affect the functionality of the OPRD1–OPRM1 heterodimer and the selection of the activated pathway.

### 2.8. Structural Insights

The mutation A6V is located at the beginning of the N-terminal of the OPRM1 protein, and it is within a flexible loop. The mutation site is partially exposed to the water phase; thus, there are no issues with a replacement of the small Ala residue with Val. Furthermore, the physicochemical property differences, as measured by a quantity referred as “protein distance” is only 0.1 (see Reference [59] for details). In addition, A6V is not located on the protein–protein interface in any of the generated models and thus does not have a significant effect on the binding free energy change.

The mutation N40D is located in a helix in the extracellular domain on the OPRM1 protein. Both amino acids Asn and Asp are small residues; however, Asn is uncharged while Asp is negative charged. The physicochemical property difference is 1.28, which is in the moderate range (see Reference [59] for details), much larger than that of A6V. As a result, N40D is expected to have a larger impact on the stability of the extracellular domain of OPRM1 than it will on the A6V mutation. Similar to the A6V mutation, the N40D mutation is close to but not directly on the protein–protein interface in any of the predicted models and thus has little effect on the binding free energy change.

The binding pockets of OPRD1 and OPRM1 are located inside the transmembrane (TM) domains between TM3, TM5, TM6, and TM7 and are close to the extracellular side [60]. Mutations A6V and N40D are on the extracellular domain but away from the binding pocket. However, while the mutations are far away from the binding pockets, the extracellular domains are situated on the top of the entrance of the binding pockets. Mutation site 6 (mutation A6V) is around 38 angstroms away from the center of the binding pocket of the OPRM1 protein in the heterodimer model we built, and mutation site 40 (mutation N40D) is around 25 angstroms from the center of the binding pocket of the OPRD1 protein. However, it is plausible that the changes in the folding free energy of the extracellular domain of OPRM1 caused by mutations could affect their structure and affect the pathway of the substrates to the binding pocket(s).

While the OPRD1–OPRM1 heterodimerization was shown to play an important role in pain control with opioid medications, the impact on addiction is not clear yet [61]. However, both mutations are predicted to decrease the N-terminal domain stability and to affect heterodimerization. It can be speculated that these changes may affect the selection of the signaling pathway [62]. Consequently, the mutant heterodimer may not have the wild-type ability to mediate the signaling pathways.

## 3. Methods

### 3.1. 3D Structure Modeling

Protein structures of OPRM1 and OPRD1 were predicted with the I-TASSER server [46]. The OPRD1–OPRM1 heterodimer structures were predicted by the ZDOCK server (V3.0.2) [47]. Both ZDOCK and the ClusPro protein-protein docking server (V2.0) were used to build the interaction model of the extracellular part of the heterodimer [50]. The connection of extracellular structures with transmembrane structure was refined by Modeller [63] through Chimera [64]. Membrane bilayers were generated through the CHARMM-GUI website (V3.1) [51]. ZDOCK was also used to predict the binding models of the OPRD1–OPRD1 heterodimer with the G-protein or β-arrestin.

### 3.2. Mapping the Mutations onto the 3D Structure of OPRM1

Mutations of A6V and N40D were generated by a mutator plugin, v1.3, in Visual Molecular Dynamics (VMD) (V1.9.3) [65].

### 3.3. Analyzing Folding Free Energy Change Due to Mutation

The extracellular domain of OPRM1 was used to calculate the folding free energy changes due to mutations A6V and N40D. They were calculated by the SAAFEC-SEQ algorithm [66], along with third party webservers such as mCSM Protein Stability Change Upon Mutation web version [67], SDM(V2) [68], DUET [57], CUPSAT [58], and I-Mutant 3.0 [69].

### 3.4. Analyzing Binding Free Energy Change Due to Mutation

Binding free energy change of OPRD1–OPRM1 complexes due to mutations A6V and N40D was computed with the in-house algorithm and the SAAMBE-3D [70] method along with third party tools such as BeAtMuSiC [71], mCSM-PPI2 [72], and MutaBind2 [73]. Only extracellular domains were used for the calculations.

## 4. Conclusions

In this work, we reported structural modeling of the OPRD1–OPRM1 heterodimer along with the G-protein or β-arrestin. In addition to providing these structural models for further investigation, the fact that the models were obtained with the same degree of confidence indicates that the G-protein and β-arrestin can bind to either OPRD1 or OPRM1 within the OPRD1–OPRM1 heterodimer. Furthermore, we examined the folding free energy and binding free energy changes due to mutations A6V and N40D within the N-terminal domain of OPRM1. We found that the folding free energy change of N40D mutations is greater than half a kcal/mol. It is speculated that this may affect the stability of the extracellular structure of the OPRM1 protein, and thus the hetero-dimerization and the selection of the signaling pathway. Both A6V and N40D mutations of OPRM1 were predicted not to have a significant effect on the binding free energy; however, it is plausible that even small changes could affect the functionality. Thus, the small changes in the folding and binding free energy predicted for each of the mutations are expected to alter the wild-type functionality of the OPRD1–OPRM1 heterodimer and, as a result, the selection of the signaling pathway. Taken together, this work provides some clues of the plausible implications of the A6V and N40D mutations on the structural integrity of the OPRM1 extracellular domain and thus on their link with opioid addiction. The structural models are expected to be used for more detailed investigations together with bound ligands to further probe the effect of mutations.

## Figures and Tables

**Figure 1 ijms-22-10290-f001:**
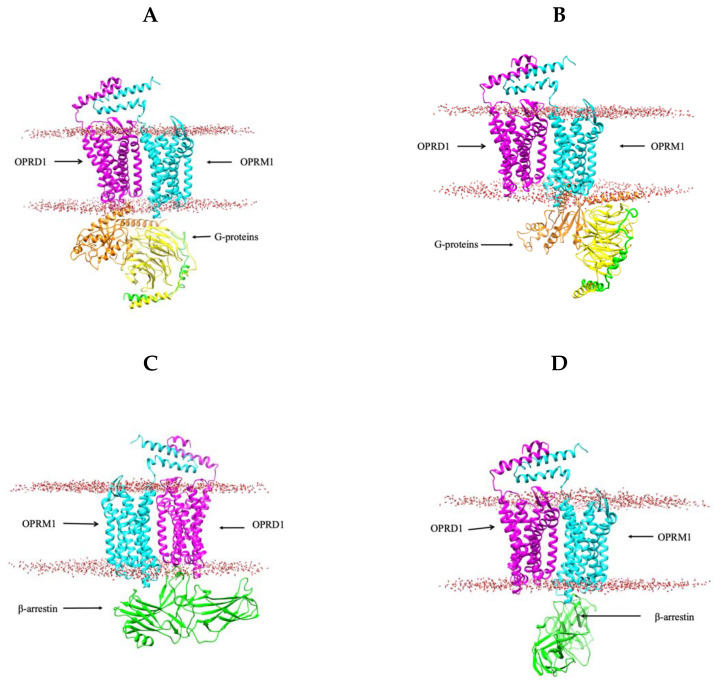
Three-dimensional structures of the OPRD1–OPRM1 heterodimer with signaling protein(s). (**A**) The G-protein complex binds to OPRD1 on the OPRD1–OPRM1 heterodimer. (**B**) The G-protein complex binds to OPRM1 on the OPRD1–OPRM1 heterodimer. (**C**) β-arrestin binds to OPRD1 on the OPRD1–OPRM1 heterodimer. (**D**) β-arrestin binds to OPRD1 on the OPRD1–OPRM1 heterodimer.

**Figure 2 ijms-22-10290-f002:**
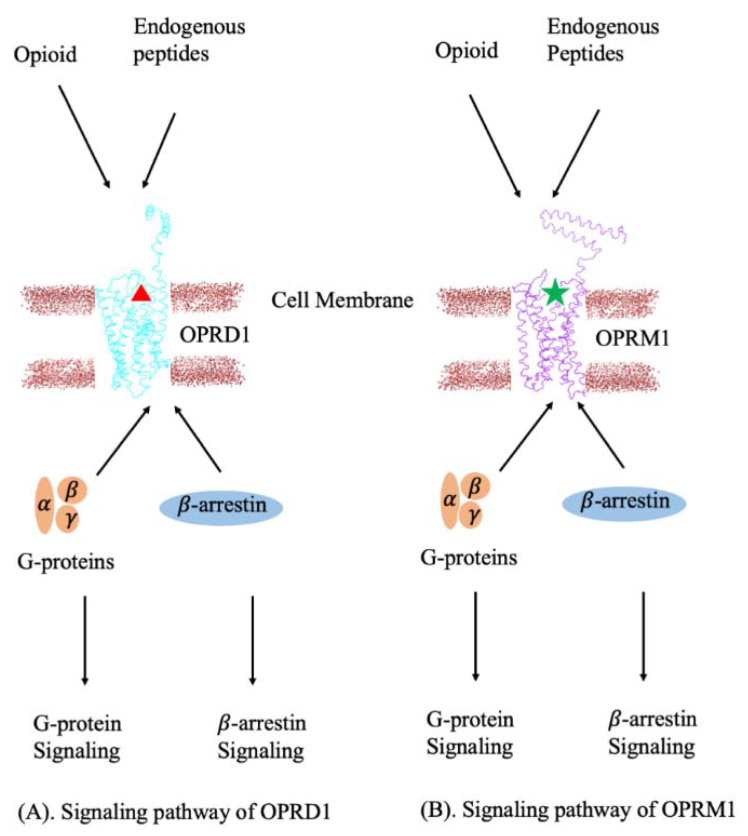
Signaling pathway of OPRD1 and OPRM1 monomers. (**A**) When an opioid ligand or an endogenous peptide binds to the binding pocket of OPRD1 protein, it will active the binding of G-proteins or β-arrestin and then activate the corresponding signaling pathway. For example, when a β-endorphin binds to OPRD1protein, the β-arrestin signaling pathway will be activated. (**B**) When an opioid ligand or an endogenous peptide binds to the binding pocket of an OPRM1 protein, it will active the binding of G-proteins or β-arrestin and then activate the corresponding signaling pathway. For example, when an enkephalin binds to OPRM1protein, the β-arrestin signaling pathway will be activated.

**Figure 3 ijms-22-10290-f003:**
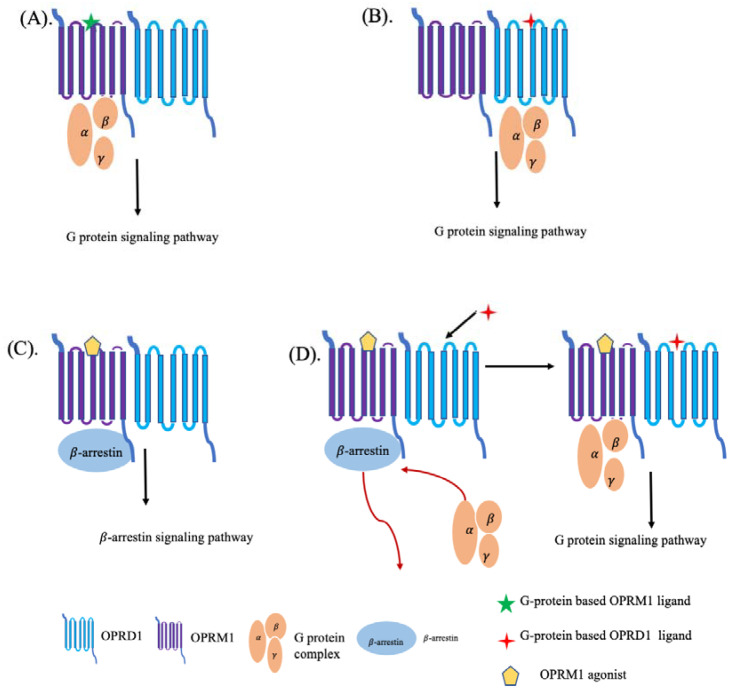
Ligand-mediated signaling pathways on the OPRD1–OPRM1 heterodimer. (**A**) When a G-protein-based OPRM1 ligand binds to the binding pocket of OPRM1 protein, it will activate the G-protein’s signaling pathway. (**B**) When a G-protein-based OPRD1 ligand binds to the binding pocket of an OPRD1 protein, it will activate the G-protein’s signaling pathway. (**C**) When an OPRM1 agonist binds to the binding pocket of an OPRM1 protein, it will activate the β-arrestin signaling pathway. (**D**) When an OPRM1 agonist has bound to the binding pocket of an OPRM1 protein while a G-protein-based OPRM1 ligand binds to the binding pocket of an OPRD1 protein, it will switch the β-arrestin signaling pathway to the G-protein’s signaling pathway.

**Table 1 ijms-22-10290-t001:** Folding free energy change due to mutations on OPRM1. Negative sign means destabilization. SD–standard deviation in kcal/mol.

	SAAFEC-SEQ	mCSM	SDM	DUET	CUPSAT	I-Mutant 2.0	Avg (kcal/mol)	SD
A6V	−0.86	−0.216	−0.24	−0.045	0.26	−0.16	−0.211	0.367
N40D	−0.73	−1.58	−0.03	−1.227	−0.86	−0.8	−0.608	0.522

**Table 2 ijms-22-10290-t002:** Binding free energy change due to mutations on OPRD1–OPRM1 heterodimer. Negative sign means destabilization. SD–standard deviation in kcal/mol.

	SAAMBE-3D	mCSM	BeAtMusic	Mutabind 2	Avg(kcal/mol)	SD
A6V	−0.01	−0.389	−0.34	−0.38	−0.27975	0.181
N40D	−0.23	−0.163	−0.28	−0.44	−0.27825	0.118

## Data Availability

All models can be downloaded from https://drive.google.com/drive/folders/1FKoNL-BtQ0j33_S8HhtSqWds79YByk4d, accessed on 29 August 2021.

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
