# Peer review of "Opioid Addiction and Opioid Receptor Dimerization: Structural Modeling of the OPRD1 and OPRM1 Heterodimer and Its Signaling Pathways"

_ijms, 2021, doi:10.3390/ijms221910290_

Round 1
Reviewer 1 Report
This manuscript demonstrated that the mutations (A6V and N40D) that have been implicated in opioid addiction may destabilize the extracellular domain of OPRM1, while it may not affect the binding free energy change.
The results of this study may include beneficial information to understand the importance of addiction-related mutations of opioid receptor on the heterodimerization, but I found several problems as following.
Specific comments
- Although authors discuss that the mutation will results in the destabilization of extracellular domain of OPRM1, there are no direct results or evidences that clearly explain how they affect the functionality of OPRD1-OPRM1 heterodimer and the selection of the activated signaling pathway that is the most important point of this study.
- Also, there are no discussion on the relationship between addiction and the results of this study. That is, are there any possible relationship between the effect of mutations on the stability of the extracellular domain of OPRM1 or heterodimer function and the addiction?
- Throughout the manuscript, since typos (such as A6T or A6V, OPRD1 or OPRM1) are found, authors should carefully re-check the manuscript. For example, in Figure2, beta-endorphin may be an agonist of MOR, and enkephalins may be an agonist of DOR, aren’t they?
Author Response
We thank the reviewer for useful suggestions and comments. They were carefully considered, and appropriate changes were made in the manuscript. Below we address reviewer’s comments point-by-point.
- Although authors discuss that the mutation will results in the destabilization of extracellular domain of OPRM1, there are no direct results or evidences that clearly explain how they affect the functionality of OPRD1-OPRM1 heterodimer and the selection of the activated signaling pathway that is the most important point of this study.
Response: Revisions were made in the last paragraphs on page 2 and page 8. Appropriate references were added to address reviewers’ questions.
- Also, there are no discussion on the relationship between addiction and the results of this study. That is, are there any possible relationship between the effect of mutations on the stability of the extracellular domain of OPRM1 or heterodimer function and the addiction?
Response: Discussion is added on the pages 8-9.
3. Throughout the manuscript, since typos (such as A6T or A6V, OPRD1 or OPRM1) are found, authors should carefully re-check the manuscript. For example, in Figure2, beta-endorphin may be an agonist of MOR, and enkephalins may be an agonist of DOR, aren’t they?
Response:
3.1. The typos of A6T were fixed. We also changed MOR and DOR in the figure 2 and figure 3 to OPRM1 and OPRD1.
3.2. Yes, beta-endorphin may be an agonist of MOR, and enkephalins may be an agonist of DOR. In the figure, those two peptides are just examples (The examples were discussed at second sentence of last paragraph on page 5), and it does not state they are only able to bind to MOR or DOR. To eliminate the confusion, they were removed from the figure 2 in the revision.
Reviewer 2 Report
Opioid addiction and opioid receptors dimerization: Structural modeling of OPRD1 and OPRM1 heterodimer and its signaling pathways
The work can conducted to investigate two mutations, A6V and N40D, found in mu opioid receptor gene OPRM1. The mutations are located in the extracellular domain of the corresponding protein, which is important to hetero-dimerization of OPRM1 with delta opioid receptor protein (OPRD1).
The work is well described and focused on four 3D structure of molecular pathways, G-protein signaling pathway and β-arrestin signaling pathway of heterodimer of OPRD1-OPRM1.
Authors also analyzed the effect of mutations of A6V and N40D on stability of individual OPRM1/OPRD1 molecules and OPRD1-OPRM1 and concluded that hetero-dimerization is a key step for signaling processes
I have some minor comments:
Abstract: the method should include the name of the software
The following Methods are well described
2.1. 3D structure modelling
2.2. Mapping the mutations onto the 3D structure of OPRM1
2.3. Analyzing Folding Free Energy Change Due to Mutation
Comment: the country, year, version of the softwares used should be reported
Results: Pls report the SD in Table 1 and 2
Conclusion:
In this work we reported structural modeling of the OPRD1-OPRM1 heterodimer 257 along with G-protein.
G-protein and arrestin can bind to either OPRD1 or OPRM1 260 within OPRD1-OPRM1 heterodimer.
Binding free energy changes due to mutations A6V and N40D within N-terminal domain of OPRM1.
We found that the folding free energy change of N40D mutations is greater than half of kcal/mol. It is speculated that may affect the stability of extracellular structure of the OPRM1 protein and thus the hetero-dimerization and selection of the signaling pathway. Both A6V and N40D mutations of OPRM1 are predicted not to have significant effect on the binding free energy, however, it is plausible that even small changes could affect the functionality. Taken together, the work provides some clue of plausible implications of A6V and N40D mutations on structural integrity of OPRM1 extracellular domain and thus on their linkage with opioid addiction. The structural models are expected to be used for more detailed investigations together with bound ligands to further probe the effect of mutation
Comment: The conclusion should focus on the main findings rather than reiteration of methods/conducted experiments
Author Response
We thank the reviewer for useful suggestions and comments. They were carefully considered, and appropriate changes were made in the manuscript. Below we address reviewer’s comments point-by-point.
The following Methods are well described
2.1. 3D structure modelling
2.2. Mapping the mutations onto the 3D structure of OPRM1
2.3. Analyzing Folding Free Energy Change Due to Mutation
Comment: the country, year, version of the softwares used should be reported
Response: We addressed reviewer’s suggestions where it was possible. Thus, some of them as I-TASSER does not have a version number. The description of other software has been updated to include their version, and other details if available.
Results: Pls report the SD in Table 1 and 2
Response: The SD results were added in Table1 and Table 2.
Conclusion:
Comment: The conclusion should focus on the main findings rather than reiteration of methods/conducted experiments
Response: Conclusion has been revised.
Round 2
Reviewer 1 Report
I think the manuscript has been sufficiently improved to warrant publication in IJMS.